# Analysis of opticin binding to collagen fibrils identifies a single binding site in the gap region and a high specificity towards thin heterotypic fibrils containing collagens II, and XI or V/XI

Uwe Hansen[1,2]*, David F. Holmes[3], Peter Bruckner[1], Paul N. Bishop [4,5]*

1 Department of Physiological Chemistry & Pathobiochemistry, University Hospital of Münster, Münster, Germany, 2 Institute of Musculoskeletal Medicine, University Hospital Münster, Münster, Germany, 3 Wellcome Trust Centre for Cell-Matrix Research, School of Biological Sciences, Faculty of Biology, Medicine and Health, University of Manchester, Manchester, United Kingdom, 4 Division of Evolution and Genomic Sciences, School of Biological Sciences, Faculty of Biology, Medicine and Health, University of Manchester, Manchester, United Kingdom, 5 Manchester Royal Eye Hospital, Manchester University NHS Foundation Trust, Manchester Academic Health Science Centre, Manchester, United Kingdom

* paul.bishop@manchester.ac.uk (PNB); uhansen@uni-muenster.de (UH)

**Data Availability Statement:** All relevant data are within the paper and its Supporting Information files.

## Abstract

Opticin is a class III member of the extracellular matrix small leucine-rich repeat protein/proteoglycan (SLRP) family found in vitreous humour and cartilage. It was first identified associated with the surface of vitreous collagen fibrils and several other SLRPs are also known to bind collagen fibrils and it some cases alter fibril morphology. The purpose of this study was to investigate the binding of opticin to the collagen II-containing fibrils found in vitreous and cartilage. Electron microscopic studies using gold labelling demonstrated that opticin binds vitreous and thin cartilage collagen fibrils specifically at a single site in the gap region of the collagen D-period corresponding to the e2 stain band; this is the first demonstration of the binding site of a class III SLRP on collagen fibrils. Opticin did not bind thick cartilage collagen fibrils from cartilage or tactoids formed *in vitro* from collagen II, but shows high specificity for thin, heterotypic collagen fibrils containing collagens II, and XI or V/XI. Vitreous collagen fibrils from opticin null and wild-type mice were compared and no difference in fibril morphology or diameter was observed. Similarly, *in vitro* fibrillogenesis experiments showed that opticin did not affect fibril formation. We propose that when opticin is bound to collagen fibrils, rather than influencing their morphology it instead hinders the binding of other molecules to the fibril surfaces and/or act as an intermediary bridge linking the collagen fibrils to other non-collagenous molecules.

**Funding:** This work was funded by the Deutsche Forschungsgemeinschaft (SFB 492 project A2 to P. B.; https://www.dfg.de/en/) and UK Medical Research Council (MRC Reference: MR/M025365/1 to P.N.B.; https://mrc.ukri.org/). The funders had no role in study design, data collection and analysis, decision to publish, or preparation of the manuscript.

**Competing interests:** The authors have declared that no competing interests exist.

## Introduction

Opticin is a member of the extracellular matrix small leucine-rich repeat protein/proteoglycan (SLRP) family that was first identified in vitreous humour and subsequently in cartilage [1,2]. There are 18 members of the SLRP family which have been divided into 5 classes based upon phylogeny [3]. Opticin is in class III and is therefore closely related to the other class III SLRPs, epiphycan and osteoglycin/mimecan. Little is known about the functions of class III SLRPs, but opticin has been shown to possess anti-angiogenic properties both *in vitro* and *in vivo*, and mechanistic experiments suggested that its anti-angiogenic effects are elicited by opticin binding to collagen resulting in steric inhibition of endothelial cell integrin-collagen interactions [4,5]. However, this conclusion was drawn from experiments using monomeric collagens, whereas *in vivo* these collagens are in fibrillar structures. Furthermore, recent work has demonstrated that integrins that bind collagen monomers do not bind collagen in fibrils directly, instead these integrins interact with non-collagenous molecules associated with the surface of the fibrils [6]. It has also been demonstrated that opticin deficient mice are protected against osteoarthritis because the lack of opticin results in an alteration in the amounts of other SLRPs in cartilage leading to altered fibril diameter and increased protection from proteolysis [7]. The above studies prompted us to investigate interactions between opticin and collagen fibrils.

The vitreous humour contains a network of fine collagen fibrils of uniform thickness (10–20 nm depending upon species) that impart gel-like properties to the tissue [8], whereas in human cartilage there are both thin ~ 18 nm diameter fibrils and thicker fibrils [9,10]. The collagen fibrils from these two tissues are very similar in their collagen composition: They are heterotypic (mixed in composition) fibrils in which fibril-forming type II collagen predominate, the fibrils also contain collagen IX and XI or V/XI. Collagen IX is a part-time proteoglycan and a member of the FACIT (Fibril-Associated Collagens with Interrupted Triple-helices) group of collagens that is located on the surface of the fibrils and has a role in preventing fibril aggregation [9,11]. Collagen XI and V/XI are very closely related fibril-forming collagens both containing an α1(XI) chain, in addition collagen XI contains α2(XI) and α3(XI) chains and vitreous collagen V/XI contains an α2(V) chain, but the identity of the third α-chain is uncertain [12]. Hybrid molecules containing α1(V) and α2(XI) collagen chains have also been described in articular cartilage [13]. The collagen content of the fibrils determines their morphology and it is possible to generate heterotypic collagen fibrils *in vitro* that have identical morphology to their tissue counterparts [14]. Natural collagen II alone does not form fibrils *in vitro* but rather forms short, tapered tactoids with D-periodic banding. However, when mixed with sufficient type XI collagen, thin fibrils of uniform diameter are formed. Here, using fibrils formed *in vivo* and *in vitro*, we investigated the binding of opticin to collagen fibrils and the effects of opticin on fibril morphology.

## Results

### Analysis of vitreous collagen fibrils from wild-type and opticin null mice

The vitreous collagen fibrils of opticin null and wild-type mice appeared morphologically identical (Fig 1A and 1B). The mean diameter of the fibrils from the opticin null mice was 9.99 nm +/- 2.12 nm (n = 286) and from wild-type mice was 10.77 nm +/- 2.17 nm (n = 290); statistical analysis revealed no significant difference (p = 0.38); this and subsequent statistical analyses used the two-sided Student's t-test.

### Opticin localisation of authentic bovine vitreous and cartilage collagen fibrils

Bovine vitreous collagen fibrils were analysed by immunoelectron microscopy using polyclonal opticin antibodies and gold-conjugated anti-rabbit immunoglobulins. These collagen

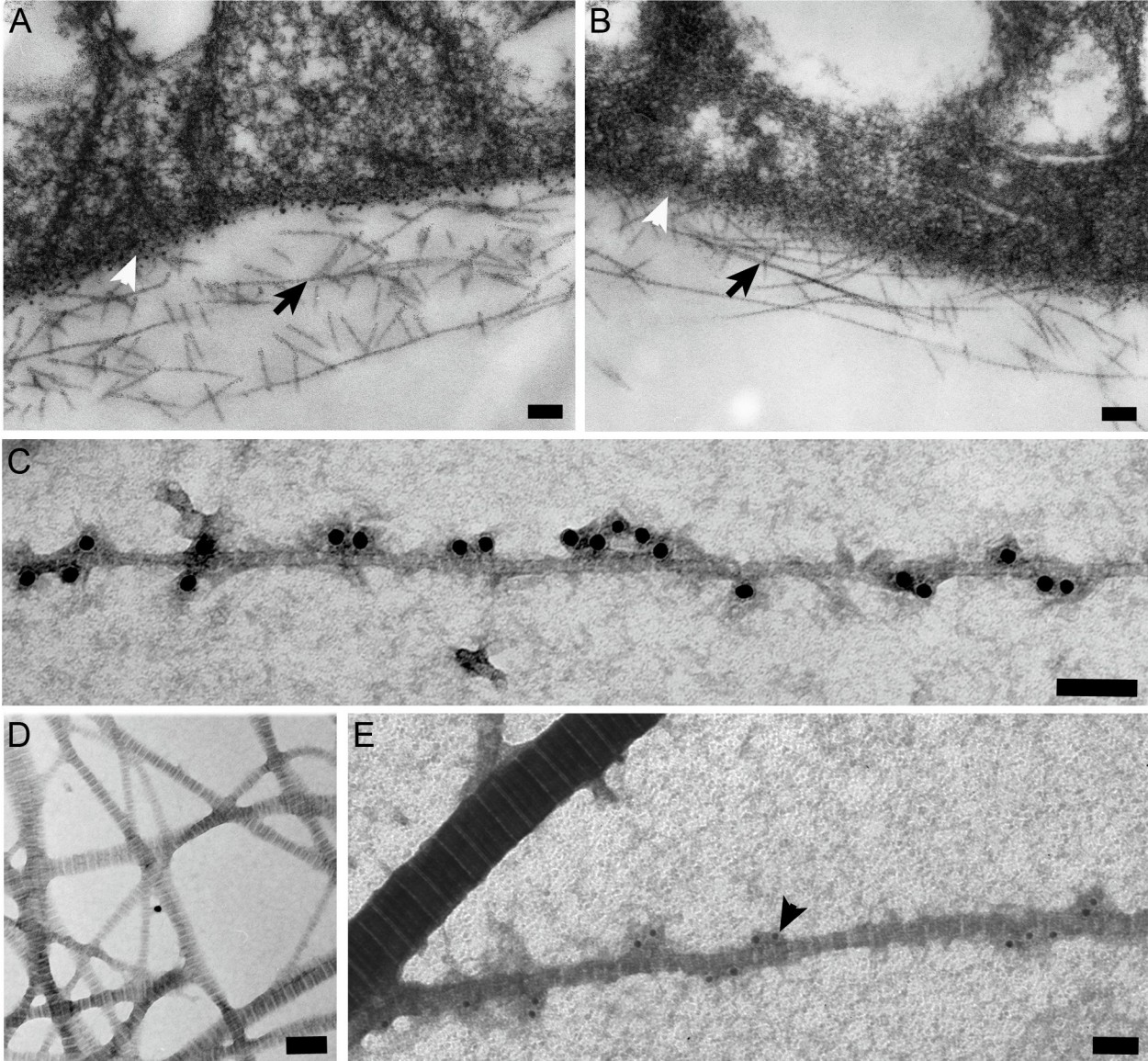

**Fig 1. Ultrastructural analyses of vitreous and cartilage collagen fibrils.** Electron microscopy of vitreous collagen fibrils (black arrows) from wild-type (A) and opticin null (B) mice showing morphological identity; there was no difference in mean fibril diameter (white arrow heads indicate inner retinal surface). (C-E) Opticin antibody in conjunction with 18 nm gold-conjugated secondary antibody demonstrates opticin on the surface of bovine vitreous collagen fibrils (C); endogenous opticin was not detected on cartilage collagen fibrils (D). Recombinant opticin added to cartilage fibrils *in vitro* was detected by immunogold labelling on the surface of thin (black arrowhead), but not thick fibrils (E, bars 100nm).

fibrils showed extensive labelling (Fig 1C). Whilst opticin has been identified in cartilage [2,7], we could not detect endogenous opticin by immunogold electron microscopy in our preparations of cartilage fibrils preparations (Fig 1D). Therefore, we investigated the localisation of opticin after it had been added to authentic cartilage fibril preparations that had been digested with chondroitin ABC lyase (to prevent steric hindrance from glycosaminoglycans). We then observed that opticin localised to the thin fibrils, but not thick collagen fibrils (Fig 1E).

## Localisation of opticin on vitreous collagen fibrils in relation to D-period

To locate precisely the site of opticin binding, vitreous collagen fibrils were incubated with recombinant opticin that had been directly labelled with gold particles and the position of binding in relation to the D-period was analysed (Fig 2A). Along the 67 nm D-period, a binding site was observed at 39 nm from the N-terminal end of the gap region (Fig 2B and 2C). When this was correlated with the banding pattern according to the notation of Schmitt and Gross [15] the peak position of opticin binding (Fig 2B) corresponded to the position of the e2 positive stain band in the gap region, S1 Fig. Comparison of this binding site to the established axial structure of the vitreous heterotypic fibrils [16], S1 Fig, showed that there are four possible sites of interaction of the opticin on the triple helices of both collagen II (Fig 2D) and collagen V/XI molecules (Fig 2F). Furthermore, interaction of the bound opticin with all 3 domains of the type IX collagen molecule is possible (Fig 2E), but not with the retained N-propeptide of the type V/IX collagen molecule (Fig 2F).

## Opticin binding to collagen monomers and effect during fibril reconstitution

We have previously shown that recombinant bovine opticin binds to immobilised bovine collagen types I and II monomers using microtiter plate assays [4]. Prior to studying opticin binding to reconstituted fibrils, we investigated the binding of opticin to monomeric chick collagens using solid-phase assays and found that it bound chick collagens II, IX and IX, S2 Fig. Collagen fibrils were generated *in vitro* using purified chick collagens II, IX, and XI as previously described [14]. Collagen fibrils were generated using either mixtures of collagens II and XI or II, IX and XI in molar proportions of 8:1 or 8:1:1, respectively. These mixtures were supplemented with 0–50 μg/ml of opticin, added at the start of fibrillogenesis. In mixtures of collagens II/IX/XI or II/XI turbidity developed without a lag phase and the shape of the turbidity curves was unaffected by the presence of opticin, S3 Fig; therefore, opticin did not affect the rate of fibril formation. Immunoelectron microscopy confirmed the presence of opticin on the surface of the reconstituted fibrils, S4 Fig. Analysis of the reconstituted fibrils demonstrated that opticin did not significantly affect the final mean diameter of the collagen II/XI fibrils (19.71 +/- 4.07 nm (n = 118) without opticin; 20.38 +/- 3.29 nm (n = 102) with 25 μg/ml of opticin, p = 0.21). In the case of collagen II/IX/XI fibrils the final mean diameter was slightly increased by the presence of 25 μg/ml opticin during fibrillogenesis (17.64 +/- 2.90 nm (n = 142) without opticin; 18.89 +/- 3.19 nm (n = 130) with opticin, p = 0.008).

## Binding of opticin to preformed fibrils and tactoids

Opticin bound to the surface of fibrils containing collagens II/XI or II/IX/XI when added after reconstitution of fibrils (Fig 3A and 3B). Scant opticin binding was also observed when added to reconstituted fibrils composed of just type XI collagen (Fig 3C and 3D). However, no binding was observed to the thick, banded tactoids formed by type II collagen alone (Fig 3E and 3F).

## Discussion

SLRPs generally bind to collagen and several have been shown to modify fibril morphology, although much of this research has been done using collagen I-containing fibrils, rather than the collagen II-containing fibrils found in vitreous and cartilage [18]. Whilst several SLRPs are present in vitreous [19], opticin is of particular interest because it was found to be a major component of a pool of macromolecules that strongly associated with vitreous collagen fibrils

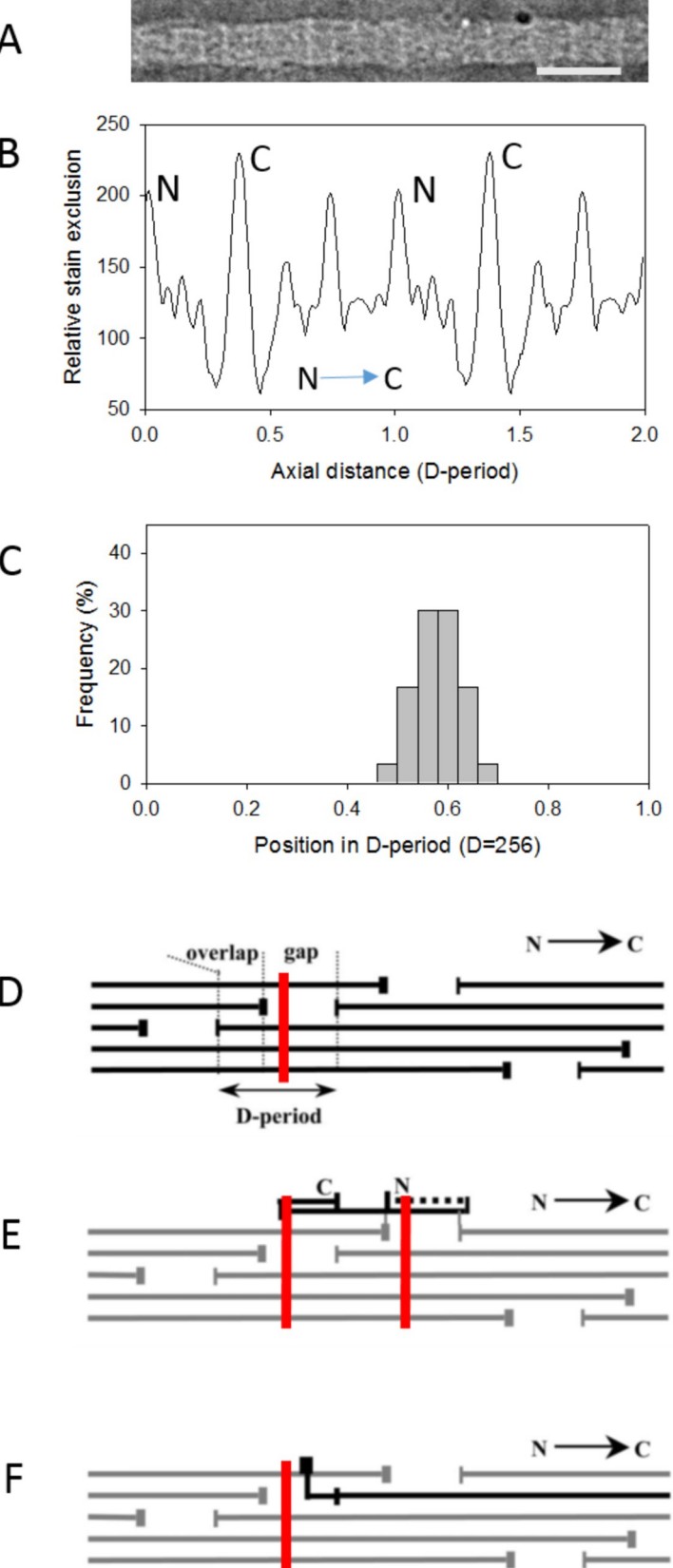

**Fig 2. Ultrastructural localization of gold-conjugated opticin.** Opticin that had been directly conjugated to 6 nm gold particles (white arrowhead) was incubated with isolated vitreous collagen fibrils and the fibrils were examined by electron microscopy (A). The average axial stain exclusion pattern (ASEP) is shown (repeated over 2 D-periods) (B); the peaks corresponding to the N- and C-terminal ends of collagen II molecules are indicated and the arrow indicates molecular polarity. This reference pattern was cross-correlated with the axial pattern from each labelled fibril to establish the molecular polarity and position of the gold label within the D-period (C). Diagrams (D-F) show the axial arrangement of collagen molecules in the heterotypic fibril producing the characteristic gap-overlap structure. Collagen II molecules are shown in black with the opticin binding position shown as a red line (D). The axial location of collagen IX [16] is shown in (E) with previously established polarity and folded back domains [17]. The axial location of the bound opticin (shown for 2 adjacent D-periods) includes all three domains of the collagen IX molecule. The axial location of collagen V/XI is shown in black (F); the retained N-propeptide protrudes onto the fibril surface and is axially located as shown. The opticin axial binding site does not overlap with the N-propeptide. The scale bar in (A) corresponds to a single D-period (67nm).

[1]. Here we confirm that opticin binds to the surface of vitreous collagen fibrils. Furthermore, opticin localises specifically to a single binding site in the gap region of the collagen fibril D-period corresponding to the e2 stain band of the collagen I (or II) fibril [20]. This is the first identification of the binding site of a class III SLRP on collagen fibrils. Other SLRPs have been shown to bind to the gap region of collagen II-containing fibrils including decorin and fibromodulin and biochemical experiments suggested that their binding sites are not overlapping [9,21,22].

Using the *in-vitro* fibrillogenesis system [14], we show that opticin has no effect on the rate of fibrillogenesis and no effect, or minimal effect, on fibril diameter. The mean diameter of fibrils containing collagens II/IX/XI was slightly greater when formed in the presence of opticin compared to when opticin was absent, but this could simply represent opticin on the fibril surfaces rather than modification to the underlying collagen structure. These fibrils formed *in vitro* closely resemble the thin cartilage collagen fibrils found *in vivo*, which have the same molar ratio of collagens and a diameter of 15–18 nm [9,10,23] and vitreous collagen fibrils which have a diameter of 10–20 nm depending upon species and method of analysis [8,16]. The conclusion that opticin does not affect fibril morphology was further supported by comparing the diameter of vitreous collagen fibrils from opticin null and wild-type mice where no significant difference was observed. It is of note that there was also no difference in the diameter of corneal collagen fibrils when mice deficient in the class III SLRP mimecan were compared to wild-type mice [24]; suggesting that generally type III SLRPs, whilst binding to collagen fibrils, do not directly influence their morphology during fibrillogenesis.

An intriguing observation was that opticin binds to the thin collagen fibrils from cartilage (that closely resemble vitreous collagen fibrils), but not the thick collagen fibrils; by contrast decorin binds to thick, but not thin cartilage fibrils [9]. To explore these observations further we investigated the binding of opticin to different types of fibrils/tactoids formed *in vitro*. Opticin did not bind tactoids formed from just type II collagen, and did not bind to fibrils just composed of just collagen I, S5 Fig, despite binding to monomeric collagens I and II. There was limited binding to fibrils composed of just collagen XI. However, once copolymers were formed from collagens II/XI or II/IX/XI resembling vitreous or thin cartilage collagen fibrils, extensive binding of opticin was observed. These data show that collagen II must form a copolymer with collagen XI (and presumably collagen V/XI in vitreous) to assemble into thin fibrils that then provide a specific opticin binding site. This is the first time that a member of the SLRP family has been shown to require a particular fibrillar collagen copolymer for binding, and this work raises the possibility that other members of the SLRP family have similarly exacting requirements for binding to collagen fibrils.

The above data suggests that the function of opticin is to modulate the surface interactions of vitreous and thin cartilage collagen fibrils rather than influence their morphology. The

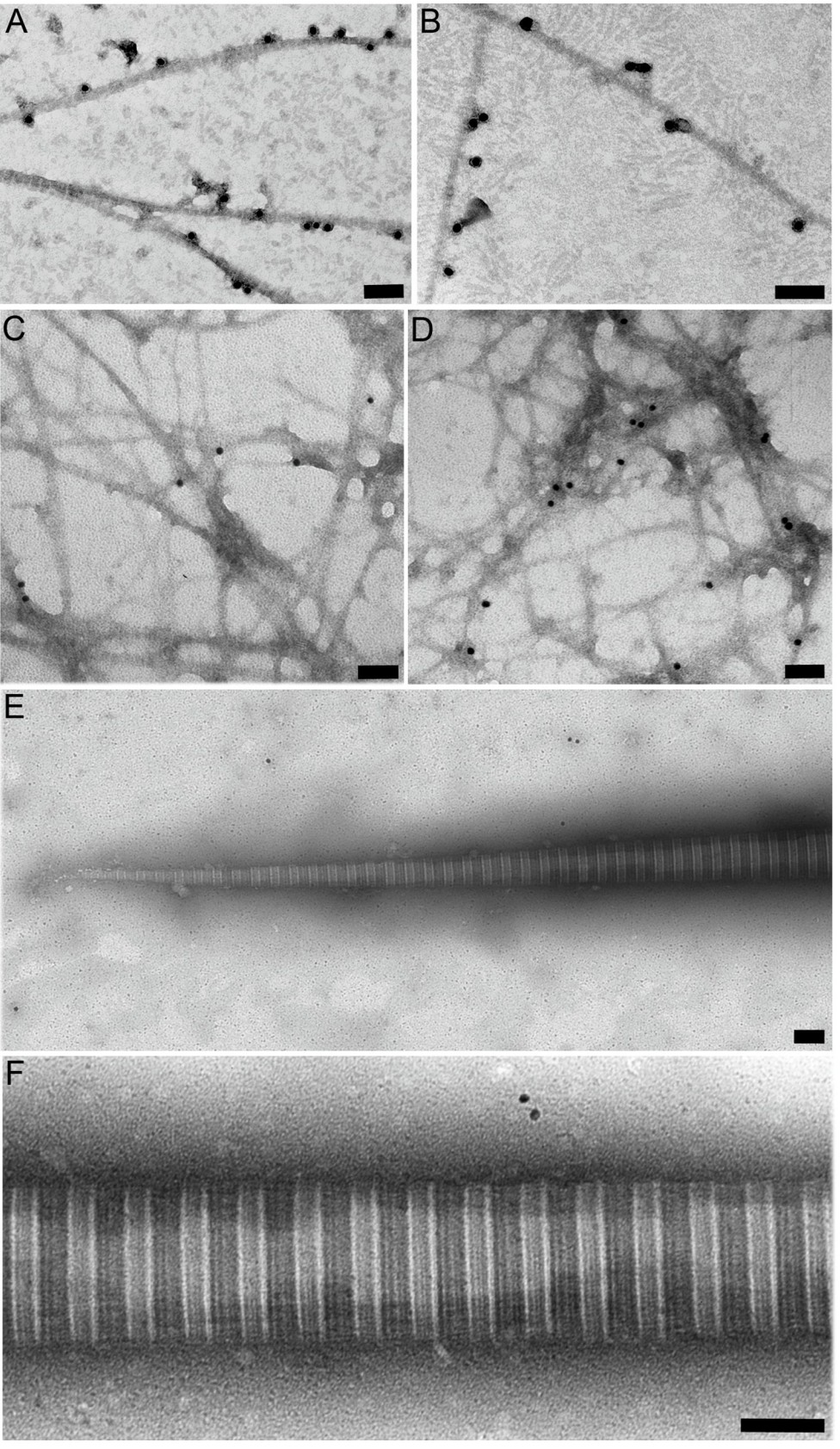

**Fig 3. Binding of opticin to pre-formed reconstituted collagen fibrils.** Fibrils were reconstituted, incubated with opticin, then analysed by immunoelectron microscopy following incubation with the opticin antibody and secondary antibody conjugated with 18 nm gold particles: (A) Fibrils containing collagens II and XI and (B) collagens II, IX and XI incubated with 10 μg/ml of opticin; collagen XI fibrils incubated with 5 (C) and 50 μg/ml (D) of opticin; (E,F) tapering end and middle part of collagen II tactoids incubated with 50 μg/ml of opticin; (bars 100 nm).

absence of opticin in vitreous promotes angiogenesis and possibly opticin prevents the binding of other non-collagenous protein to the fibril surfaces that provide endothelial cell integrin binding sites and thereby promote angiogenesis. Opticin null mice showed protection against osteoarthritis and it was found that these mice had increased levels of lumican and epiphycan, and decreased levels of fibromodulin [7]. Here opticin may prevent the fibrils from interacting with other SLRPs that do alter fibril morphology and provide increased protection against proteases. So, in conclusion, we propose that opticin has exacting requirements for collagen fibril binding and only binds a specific site on the surface of thin fibrils containing collagens II and XI or V/XI, there modulating the interactions of other molecules with the collagen surfaces.

# Materials and methods

## Materials

The production and characterisation of opticin null mice has been described previously [5], and was undertaken under institutional and UK Home Office guidelines under Project License 40/2819. Fresh bovine tissue was obtained from a local slaughterhouse; vitreous gels were isolated from the eyes and articular cartilage was prepared for electron microscopy from tarsometatarsal joints (2-yr-old cows) as described previously [25]. The production, purification and characterization of recombinant bovine opticin has been described elsewhere [26,27]. Opticin antibodies derived by affinity purification of a polyclonal rabbit antiserum raised against bovine opticin has been described previously [4].

## Electron microscopy of vitreous from opticin null and wild-type mice

Eyes from adult (less than 6 months old) opticin null mice and wild type C57BL/6 mice were fixed in 2.5% glutaraldehyde in 0.1 M sodium cacodylate buffer, pH 7.4, at room temperature. The eyes were then dehydrated in an ascending ethanol series then incubated in propylene oxide prior to embedding in Taab resin (Taab Laboratories Equipment, Aldermaston, UK). Ultrathin sections were cut and stained with 2% uranyl acetate and Reynolds lead citrate. Electron micrographs were taken at 60 kV with a Philips EM 301 electron microscope.

## Solid phase binding assays

Polystyrene microtiter plates were coated overnight at 4˚C with 10 μg of purified chick collagen II, IX or XI, or 1 μg heparin albumin (positive control). After washing with TBS, the wells were blocked by incubating with 2% (w/v) bovine serum albumin in TBS at room temperature prior to further washing with TBS containing 0.04% Tween 20 (TBS-T). Opticin was then added at a concentration ranging between 1 μg/ml to 50 μg/ml dissolved in 100 μl of TBS. After incubating for 1 h at room temperature the wells were washed with TBS-T. Bound opticin was subsequently detected with the opticin antibody followed by a horseradish peroxidase-conjugated antibody raised against rabbit immunoglobulins. Absorbance at 490 nm was measured using a microplate reader.

### *In Vitro* fibrillogenesis

Collagens II, IX, and XI from cultures of chick embryo sternal chondrocytes in agarose gels were purified in their native and fibrillogenesis-competent states as previously described [14]. Mixtures of collagens II and XI or collagens II, IX, and XI in were dissolved 0.1 M Tris-HCl, 0.4 M NaCl, pH 7.4 (storage buffer) at molar ratios of 8:1 or 8:1:1, respectively, and were reconstituted *in vitro* into fibrils. Pure collagens I, XI or II in storage buffer were also used for *in vitro* fibrillogenesis or for the generation of tactoids, respectively. Recombinant opticin was added at concentrations ranging from 0 to 50 μg/ml in storage buffer. Fibrillogenesis was initiated by dilution of the reaction mixtures with an equal volume of distilled water followed by immediate warming to 37 $^{o}$C. Fibrillogenesis was monitored by turbidity development at 313 nm and by immunogold electron microscopy.

### Immunoelectron microscopy using gold-labelled secondary antibody

Preparations containing vitreous collagen fibrils, cartilage collagen fibrils or aliquots of reconstitution products following *in vitro* fibrillogenesis were spotted onto sheets of Parafilm. Nickel grids covered with Formvar and coated with carbon were floated on the drops for 10 min to allow adsorption of material, then washed with PBS prior to treating for 30 min with PBS containing 2% (w/v) milk powder (milk solution). In some experiments, fibrils were then incubated for 1 h with exogenous recombinant opticin (concentration ranging from 0 μg/ml to 50 μg/ml) in PBS or phosphate buffer, pH 7.4, containing 1 M NaCl prior to washing several times with PBS. Next, the grids (with or without incubation with recombinant opticin) were incubated for 2 h with polyclonal opticin antibodies diluted 1:600 in milk solution. After washing five times with PBS, the grids were put on drops of milk solution containing (1:100) goat anti-rabbit IgG coated conjugated with 18 nm colloidal gold particles (Jackson ImmunoResearch). Finally, the grids were washed with distilled water and negatively stained with 2% uranyl acetate for 7 min. Control experiments were without primary antibodies. Electron micrographs were taken at 60 kV with a Philips EM 410 electron microscope.

### Electron microscopy with gold-conjugated opticin

Recombinant opticin was incubated with 6 nm gold sol following manufacturer's instructions (Aurion). Isolated vitreous collagen fibrils were incubated with opticin gold conjugate for 2 h at room temperature prior to absorbing onto carbon coated nickel grids. The grids were washed several times with distilled water and negatively stained with 5% uranyl acetate for 12 min. Electron micrographs were taken at 120 kV with an FEI Tecnai transmission electron microscope. Densitometric scanning of the D-period was undertaken as described previously [16] and the position of gold particles along the D-period was analysed. The polarity of each fibril was determined by cross-correlation with a reference average axial stain exclusion pattern (ASEP) and this also allowed the position of the gold particle label within the D-period to be determined.

## Supporting information

**S1 Fig. Relationship of the negative stain pattern of vitreous collagen fibrils to the positive and negative stain patterns of collagen I.** (A) and (B) show the positive and negative stain patterns, respectively, of collagen I fibrils. The 12 stain lines of the positive stain pattern (A) are labelled with the standard nomenclature [15]. The gap-overlap structure is indicated on the negative stain pattern [20] along with the axial locations of the N- and C-ends of the collagen molecules (B); the molecular polarity is indicated by the arrow. (C) shows the average

negative stain pattern for the vitreous fibrils in the present study oriented and aligned with the collagen I stain patterns.
(TIF)

**S2 Fig. Solid-phase binding experiments with immobilised monomeric chick collagens.** Opticin showed concentration-dependent binding to collagens II, IX and XI (heparin albumin was used as a positive control).
(TIF)

**S3 Fig. Kinetics of *in-vitro* fibril formation.** Turbidity measurements at 313 nm during fibrillogenesis with collagens II and XI or collagens II, IX and XI in the presence of varying concentrations of opticin.
(TIF)

**S4 Fig. Ultrastructure of fibrils formed *in vitro* in the presence of varying concentrations of opticin.** Collagen fibrils reconstituted from mixtures of collagens II and XI (A,C,E) and collagens II, IX, and XI (B,D,F) followed by immunoelectron microscopy with opticin antibody and gold conjugated secondary antibody. There was no labelling in the absence of opticin (A, B); when the fibrils were reconstituted in the presence of 5 μg/ml of opticin, immunogold labelling was observed (C,D), and increased labelling was observed when reconstituted with 25 μg/ml of opticin (E,F), (bars 100 nm).
(TIF)

**S5 Fig. Immunogold electron microscopy showing lack of binding of opticin to pre-formed reconstituted collagen 1 fibrils.** Collagen I was purified in a native and fibrillogenesis-competent form from tarso-metatarsal tendons of 17-day-old chicken embryos as described previously [1]. Fibrils were formed *in vitro* then immunoelectron microscopy using opticin antibodies and gold-labelled secondary antibody was performed as described in Materials and Methods. Fibrils were incubated with 5 μg/ml of opticin (A), 50 μg/ml of opticin (B) or in control experiments an equal volume of storage buffer without opticin (C), (bars 200 nm).
(TIF)

**S1 Text.**
(DOCX)

## Acknowledgments

We thank Dr Carolyn Jones (University of Manchester) for undertaking the electron microscopy on thin sections of mouse eyes.

## Author Contributions

**Conceptualization:** Uwe Hansen, Paul N. Bishop.

**Formal analysis:** Uwe Hansen, David F. Holmes.

**Funding acquisition:** Peter Bruckner, Paul N. Bishop.

**Investigation:** Uwe Hansen, David F. Holmes.

**Methodology:** Uwe Hansen, Paul N. Bishop.

**Project administration:** Paul N. Bishop.

**Writing – original draft:** Uwe Hansen, David F. Holmes, Paul N. Bishop.

**Writing – review & editing:** Peter Bruckner.

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
