## [Decision Letter · Decision Letter 0]

10 Jul 2020

PONE-D-20-16285

Analysis of opticin binding to collagen fibrils identifies a single binding site in the gap region and a high specificity towards thin heterotypic fibrils containing collagens II, and XI or V/XI.

PLOS ONE

Dear Dr. Bishop,

Thank you for submitting your manuscript to PLOS ONE. After careful consideration, we feel that it has merit but does not fully meet PLOS ONE’s publication criteria as it currently stands. Therefore, we invite you to submit a revised version of the manuscript that addresses the points raised during the review process.

Please correct minor typographic errors and shortly disccussed why you focused on opticin binding.

We look forward to receiving your revised manuscript.

Kind regards,

Dragana Nikitovic, Ph.D

Academic Editor

PLOS ONE

Journal Requirements:

Reviewers' comments:

Reviewer's Responses to Questions

**Comments to the Author**

1. Is the manuscript technically sound, and do the data support the conclusions?

Reviewer #1: Yes

2. Has the statistical analysis been performed appropriately and rigorously? 

Reviewer #1: I Don't Know

3. Have the authors made all data underlying the findings in their manuscript fully available?

Reviewer #1: Yes

4. Is the manuscript presented in an intelligible fashion and written in standard English?

Reviewer #1: Yes

5. Review Comments to the Author

Reviewer #1: Congratulations to the authors for the work on this subject.

The authors used electron mircroscopic studies in order to describe opticin binding to fibrillary collagen structures in vitreous and cartilage collagen.

Their comments on opticin binding non altering morphology of the fibrillary structures are fascinating and based om measurements of fibril diameter, but is this sufficient to conclude on morphology?

Opticin is present in vitreous and cartilage but so is e.g. biglycan, so why did the authors choose to address opticin binding only?

Please correct "predominates" to "predominate" on line 64 and "collagen II, IX and IX" to "collagen II, IX and XI" on line 146

6. PLOS authors have the option to publish the peer review history of their article (what does this mean?). If published, this will include your full peer review and any attached files.

Reviewer #1: No

---

## [Author Response · Author response to Decision Letter 0]

20 Jul 2020

Dear Dr Nikitovic

Thank you for your Email concerning our manuscript. You asked us to submit a revised version on the manuscript addressing the points that were raised during the review process. We hope that these have been addressed in the revised manuscript. Below is our response to the specific issues that have ben raised.

These have been checked and I hope they meet with the PLOS ONE style requirement.

2. We note that you have included the phrase “data not shown” in your manuscript. 

We have removed “data not shown” from line 209 and added in a Supplementary Figure (Fig S5) showing that opticin does not bind to type 1 collagen fibrils.

Responses to reviewer’s comments

Their comments on opticin binding non altering morphology of the fibrillary structures are fascinating and based om measurements of fibril diameter, but is this sufficient to conclude on morphology?

Our previous work has shown that fibril diameter, morphology and molecular composition are interdependent (e.g. Blanschke et al. JBC 2000: 275: 10370-787), so we believe that conclusions can be made on morphology if opticin does not affect fibril diameter.

Opticin is present in vitreous and cartilage but so is e.g. biglycan, so why did the authors choose to address opticin binding only?

Whilst several members of the small leucine rich repeat proteoglycan (SLRP) family have been identified in vitreous and cartilage, we were particularly interested in studying opticin because in vitreous it appeared to be the major macromolecule associated with the collagen fibrils (Reardon et al 2000). Furthermore, we have shown that in vitreous opticin has anti-angiogenic properties which appear to relate to its collagen binding, so it was deemed important to investigate the binding of opticin to collagen fibrils. An additional reason is that, whilst there is previously published data showing the binding sites of class I and class II SLRPs on collagen fibrils, there is no published data on the binding of class III SLRPs. 

To emphasise the first of these points (the others have already made in the text), we have added a sentence to the discussion, sentence 2, lines 179 -182, and a reference to a vitreous proteomics study which identified several SLRPs in vitreous.

“Whilst several SLRPs are present in vitreous [19], opticin is of particular interest because it was found to be a major component of a pool of macromolecules that strongly associated with vitreous collagen fibrils [1].”

Please correct "predominates" to "predominate" on line 64 and "collagen II, IX and IX" to "collagen II, IX and XI" on line 146

Predominates has been changed to predominate

We have not made any changes to line 146 as it is not clear what the reviewer would like us to do and as far as we can see it reads satisfactorily.

Some other minor corrections have been made to the text including incorrect reference numbers (these are highlighted).

Yours sincerely

Paul Bishop

---

## [Editor Report · Decision Letter 1]

24 Jul 2020

Analysis of opticin binding to collagen fibrils identifies a single binding site in the gap region and a high specificity towards thin heterotypic fibrils containing collagens II, and XI or V/XI.

PONE-D-20-16285R1

Dear Dr. Bishop,

We’re pleased to inform you that your manuscript has been judged scientifically suitable for publication and will be formally accepted for publication once it meets all outstanding technical requirements.

Kind regards,

Dragana Nikitovic, Ph.D

Academic Editor

PLOS ONE
---

## [Editor Report · Acceptance letter]

29 Jul 2020

PONE-D-20-16285R1 

Analysis of opticin binding to collagen fibrils identifies a single binding site in the gap region and a high specificity towards thin heterotypic fibrils containing collagens II, and XI or V/XI. 

Dear Dr. Bishop:

I'm pleased to inform you that your manuscript has been deemed suitable for publication in PLOS ONE. Congratulations! Your manuscript is now with our production department. 

Kind regards, 

on behalf of

Dr. Dragana Nikitovic 

Academic Editor

PLOS ONE